# Archerfish number discrimination

**Davide Potrich\*, Mirko Zanon, Giorgio Vallortigara\***

Center for Mind/Brain Sciences, University of Trento, Rovereto, Italy

**Abstract** Debates have arisen as to whether non-human animals actually can learn abstract non-symbolic numerousness or whether they always rely on some continuous physical aspect of the stimuli, covarying with number. Here, we investigated archerfish (*Toxotes jaculatrix*) non-symbolic numerical discrimination with accurate control for covarying continuous physical stimulus attributes. Archerfish were trained to select one of two groups of black dots (Exp. 1: 3 vs 6 elements; Exp. 2: 2 vs 3 elements); these were controlled for several combinations of physical variables (elements' size, overall area, overall perimeter, density, and sparsity), ensuring that only numerical information was available. Generalization tests with novel numerical comparisons (2 vs 3, 5 vs 8, and 6 vs 9 in Exp. 1; 3 vs 4, 3 vs 6 in Exp. 2) revealed choice for the largest or smallest numerical group according to the relative number that was rewarded at training. None of the continuous physical variables, including spatial frequency, were affecting archerfish performance. Results provide evidence that archerfish spontaneously use abstract relative numerical information for both small and large numbers when only numerical cues are available.

## Editor's evaluation

This is a very informative and nicely controlled study on numerical competence in the archerfish.

**\*For correspondence:**
davide.potrich@unitn.it (DP);
giorgio.vallortigara@unitn.it (GV)

**Competing interest:** The authors declare that no competing interests exist.

## Introduction

Non-symbolic numerical estimation is an important and well-studied cognitive ability that allows humans and other animals to interact successfully with their surroundings. The development of a 'sense of number' is associated with fundamental biological needs that in many ecological contexts allow animals to estimate how many companions or enemies are around, or how much food is present in different patches – all important information to maximize fitness and reproductive success in the wild (*Nieder, 2020*).

Typically, in order to assess numerical abilities, animals are requested to discriminate between sets of visual stimuli differing in numerosity (review in *Agrillo and Bisazza, 2014*). This can be done using spontaneous attractive natural stimuli such as food or social companion, taking advantage of the animals' natural and spontaneous tendency in some ecological contexts to 'go for more'. Alternatively, operant conditioning procedures can be used that associate a particular set of stimuli with a reward. Extensive evidence supports the use of numerical information in non-human primates (e.g. *Anderson et al., 2005*; *Beran et al., 2008*; *Beran and Beran, 2004*; *Cantlon and Brannon, 2007*; *Smith et al., 2003*), as well as in other mammals (e.g. *Abramson et al., 2013*; *Benson-Amram et al., 2011*; *McComb et al., 1994*; *Perdue et al., 2012*; *Vonk and Beran, 2012*; *West and Young, 2002*), in birds (e.g. *Bogale et al., 2014*; *Ditz and Nieder, 2016*; *Garland et al., 2012*; *Pepperberg, 2006*; *Rugani et al., 2013*; *Scarf and Colombo, 2011*), in amphibians (e.g. *Krusche et al., 2010*; *Stancher et al., 2015*), in reptiles (e.g. *Gazzola et al., 2018*; *Miletto Petrazzini et al., 2018*), in fish (e.g. *Gómez-Laplaza et al., 2018*; *Potrich et al., 2019*; *Stancher et al., 2013*), and in arthropods (e.g. *Dacke and Srinivasan, 2008*; *Gross et al., 2009*; *Nelson and Jackson, 2012*; *Rodríguez et al., 2015*) (see for general reviews in vertebrates *Nieder, 2020*; *Vallortigara, 2017*; *Bortot et al., 2021*).

Numerical discrimination seems to be supported by an 'Approximate Number System' (ANS, *Butterworth, 1999*; *Nieder and Dehaene, 2009*), which discriminative accuracy is ratio dependent in accordance with Weber's law (as the ratio between two numerosity increases, the discrimination gets more difficult). Besides the ANS, an attentional working memory-based system has been claimed for by some authors as providing precise representation of small numbers (up to 3–4), the so-called 'Object Tracking System' (OTS; *Trick and Pylyshyn, 1994*), though its generality for non-human animals is debated (discussion in *Vallortigara, 2017*; *Vallortigara, 2014*).

Studies investigating the neural basis of number representation revealed selectivity of neuronal response in some areas of the brain, such as the parietal and prefrontal cortex in humans (*Kutter et al., 2018*; *Piazza et al., 2004*) and in monkeys (*Nieder et al., 2002*; *Nieder and Merten, 2007*), the nidopallium caudolaterale in crows (*Ditz and Nieder, 2016*; *Ditz and Nieder, 2015*) and the most caudal dorsal-central part of the pallium in zebrafish (*Messina et al., 2020*; *Messina et al., 2021a*) (see also reviews in *Lorenzi et al., 2021*; *Messina et al., 2021b*), suggesting that common selective pressures led to convergent evolution of numerical representation in different species (*Nieder, 2021*; *Vallortigara, 2021*).

However, one issue in all these experiments is that animals are dealing with sets of physical elements, and thus numerical information is intrinsically melted with other non-numerical properties of the stimulus, such as the area, the density, the spatial frequency, or the elements' arrangement (*Leibovich et al., 2017*). Recently, some debates have arisen concerning whether bees use abstract numerical information or rather rely on sensory properties of the stimulus for discrimination (*Howard et al., 2018*; *MaBouDi et al., 2021*).

Taking advantage of the fact that we recently developed a sophisticated script for the automatic generation of visual stimuli that can allow proper randomization and control of continuous physical variables in number sense experiments (*Zanon et al., 2021*), we decided to perform some very precisely controlled experiments to check whether fish do use number as abstract property.

We selected archerfish (*Toxotes jaculatrix*) for our study. These fish are well known for their particular hunting strategy, which consists of spitting at preys above the water surface with a precise jet of water thrown with the mouth. This attacking repertoire makes it very easy to train them to hit targets using operant conditioning (see e.g. *Newport and Schuster, 2020*).

Still, to date, no studies in archerfish have explicitly investigated abstract numerical abilities. *Leibovich-Raveh et al., 2021* showed that when archerfish make magnitude-related decisions, their choice is influenced by the non-numerical variables that positively correlate with numerosity; for instance, when exposed to two groups of dots differing in number and continuous physical information, archerfish spontaneously selected the group containing the larger non-numerical magnitudes and smaller numerosity, switching to the larger numerical set when positively correlated with all the non-numerical magnitudes.

Related to magnitude discrimination, archerfish also proved to be able to associate different geometric shapes with different food quantities (*Karoubi et al., 2017*); this would support the existence of a system dealing with magnitudes, although a specific role of numerical information remains unclear.

In our study, archerfish were trained to select one of two arrays, involving either a small and a large numerosity (Exp. 1: 3 vs 6 elements) or small numerosities only (Exp. 2: 2 vs 3 elements). After reaching a learning criterion, archerfish were tested with novel numerical comparison (2 vs 3, 5 vs 8, and 6 vs 9 in Exp. 1; 3 vs 4, 3 vs 6 in Exp. 2) to check whether the rule they used in the training phase was based on a relative judgement (select the 'largest' or 'smallest' group) or on an absolute judgement (select a specific number of items). The numerosities used were justified by the aim to investigate how fish deal with discriminating numbers that could be supported by the ANS (large numbers, >4) and the OTS (small numbers, ≤4). Previous studies showing the use of relative rules in fish employed only large numerosities (*Miletto Petrazzini et al., 2016*; *Miletto Petrazzini et al., 2015*), leaving open the question whether the same rule would be engaged even with comparisons among small numbers. All of the different continuous physical variables such as radius, total area, total perimeter, convex-hull, and inter-distance were carefully controlled for and alternately balanced across trials, ensuring that the animals could not rely on them to perform their judgement (*Figure 1*; see also Methods section for a detailed explanation of the randomization and experimental protocol). Furthermore, a statistical analysis was run for a posteriori evaluation of whether

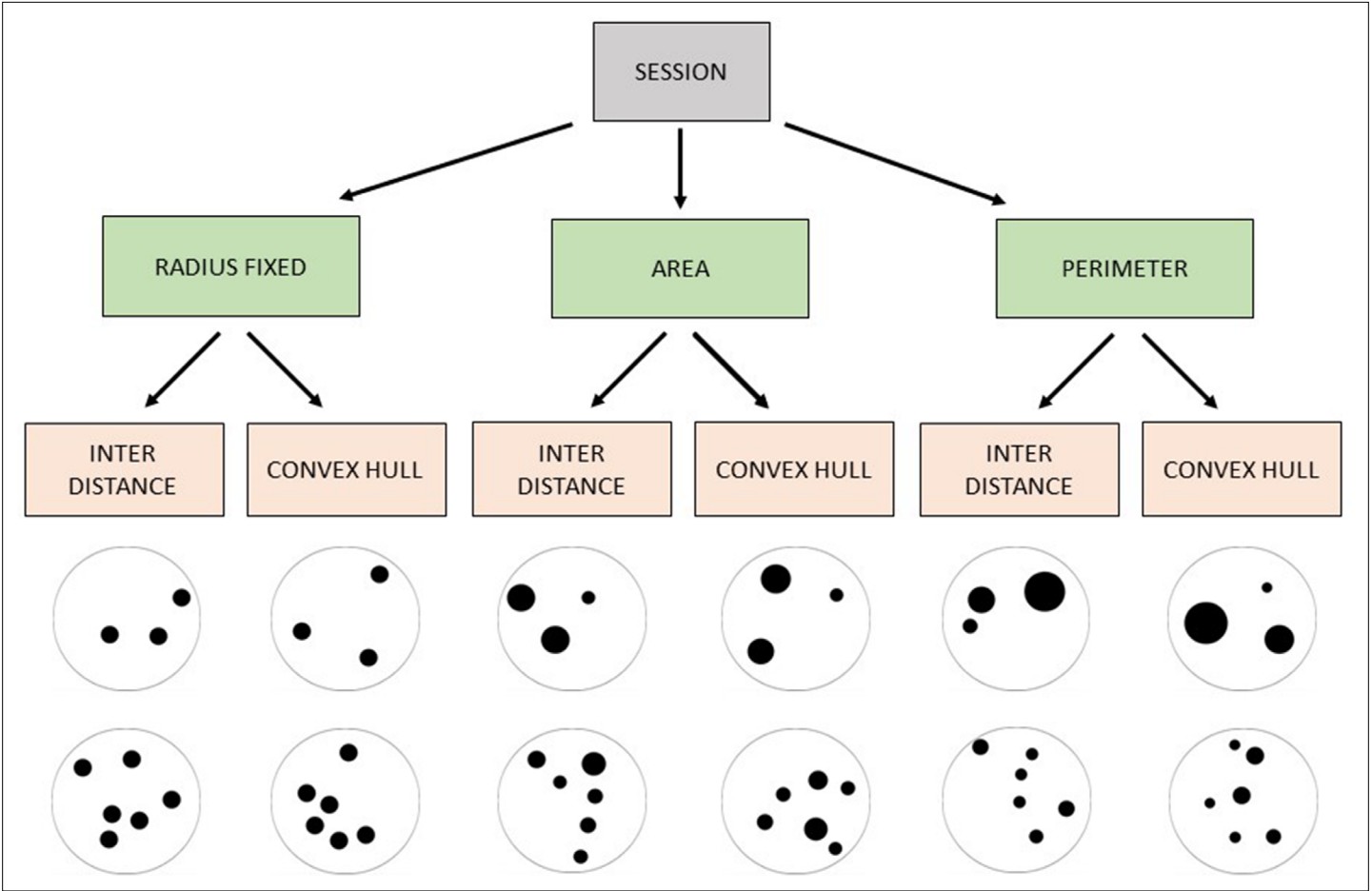

**Figure 1.** Schematic representation of the non-numerical physical controls applied to the stimuli in each session.

any of these variables influence the archerfish responses, confirming that they were not used as a cue for numerical evaluation.

## Results

### Experiment 1

Eight archerfish were trained to discriminate between two groups of black dots in a 3 vs 6 numerical comparison; four fish were trained to select the number 3, while the other four were rewarded with the number 6. Learning curves for each individual animal are reported in *Figure 2*. No difference has been found in the number of trials needed to reach the learning criterion between the group trained with three elements (mean ± standard error of the mean [SEM] = 451.25 ± 106.77) and the group trained with six elements (mean ± SEM = 413.25 ± 73.14) (independent samples *t*-test: $t(6) = 0.294$, $p = 0.779$).

When the learning criterion was reached (at least 75% of correct choices for two consecutive sessions), an analysis focused on evaluating whether archerfish' performance was influenced by the different non-numerical control conditions (i.e. overall area, overall perimeter, elements radius, elements' convex-hull, and inter-distance) was performed (see *Figure 2—figure supplement 1* for individual fish' performance in each control condition). Choices in the last two sessions (over criterion) were analyzed using a 'generalized linear mixed model' (GLMM, see Methods section). Three fixed effects (type of training – 3 dots, 6 dots; type of geometrical control – radius fixed, overall area controlled, overall perimeter controlled; type of spatial disposition control – inter-distance controlled, convex-hull controlled) and one random intercept effect (fish ID) were considered; the independent variable was the choice for the reinforced numerosity. Analysis of the random effect showed not to

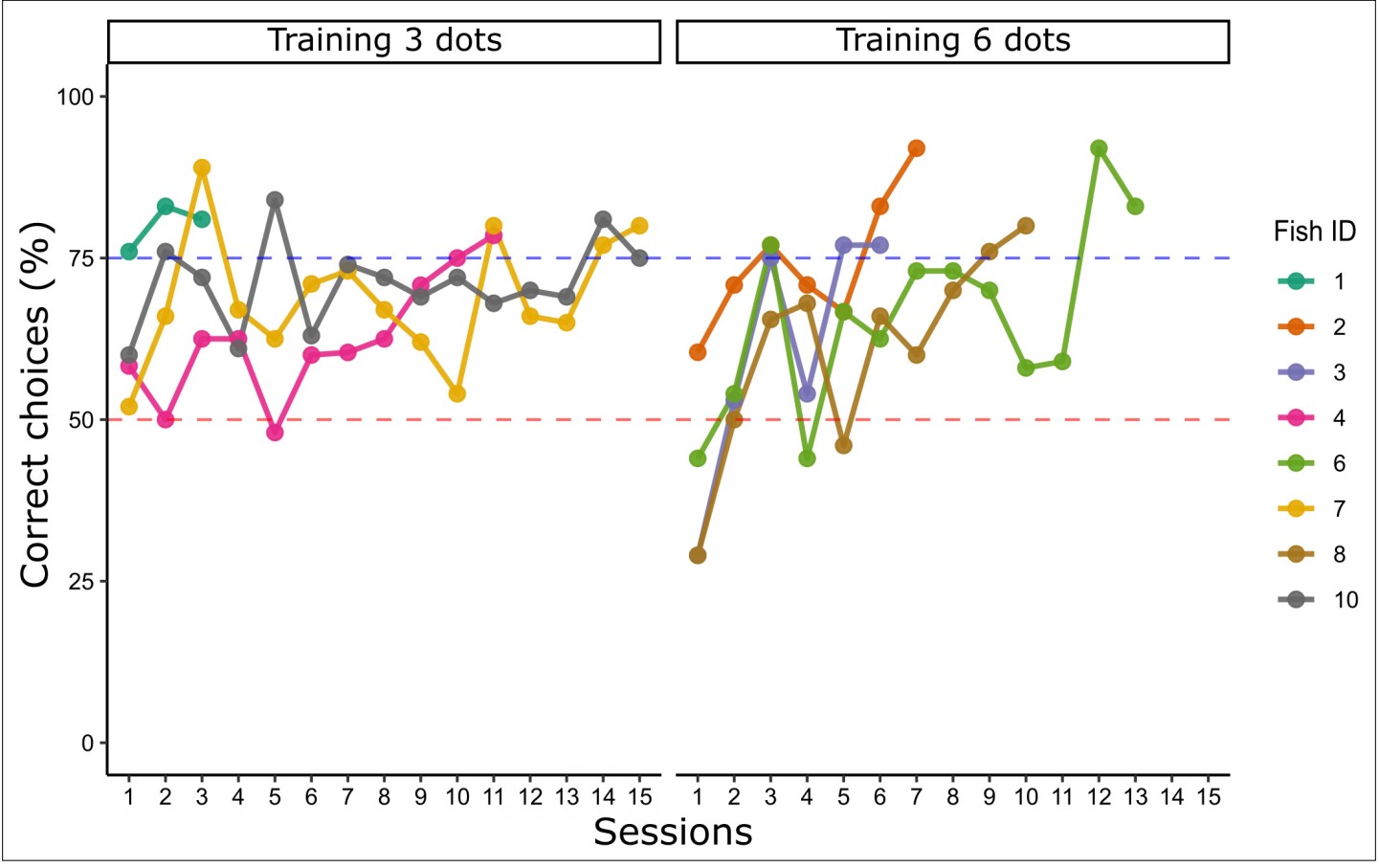

**Figure 2.** Learning curve of Experiment 1: lines graph show the percentage of correct choices for each archerfish in a 3 vs 6 training, grouped by numerosity rewarded (three or six dots). Learning criterion (blue dotted line) was reached after two consecutive sessions ≥75%. The red dotted line refers to chance level.

The online version of this article includes the following figure supplement(s) for figure 2:

**Figure supplement 1.** Performance data for each non-numerical control condition at training in Experiment 1.

affect the model (random intercept variance of the best fit: $1.14 \times 10^{-12}$). No significant differences were found between effects of groups, nor group interactions, suggesting to adopt the simplest model considering the choices with no contribution from any effect. The best model final estimate (logarithmic odds ratio) was $1.510 \pm 0.097$, corresponding to 0.82 in natural units (fraction of choices) as confirmed by a binomial test (probability of success: 81.9%, $p < 0.001$, 95% confidence intervals (CIs): 78.9–84.6). The corresponding Cohen's $g$ (see Methods section) was 0.32, indicating a large effect size.

Once the learning criterion was reached, all the fish performed three different tests.

Test 1: This test was the main discriminator to understand whether at training fish represented numerosity as relative or absolute. Fish trained to select the smallest number 3 at training (i.e. the smallest set in the 3 vs 6) were presented at test with a novel discrimination 2 vs 3, while fish trained to select the number 6 at training (i.e. largest set in the 3 vs 6) were tested with a 6 vs 9 condition. The use of 'relative' information (go for the smallest or largest) should lead the fish to choose the novel numerosity at test, while the use of 'absolute' information would reflect in the choice of the stimulus with the same number of elements as at training.

Test 2: The second test aimed to clarify the role of the incorrect (i.e. unrewarded) training stimulus and its relevance for the fish. When fish are trained to select the numerosity 3, thus avoiding number 6, once presented with the new comparison 6 vs 9 (or vice versa 2 vs 3, if trained to select 6), do they choose the group according to the relative information even if it coincides with the absolute numerosity to avoid at training?

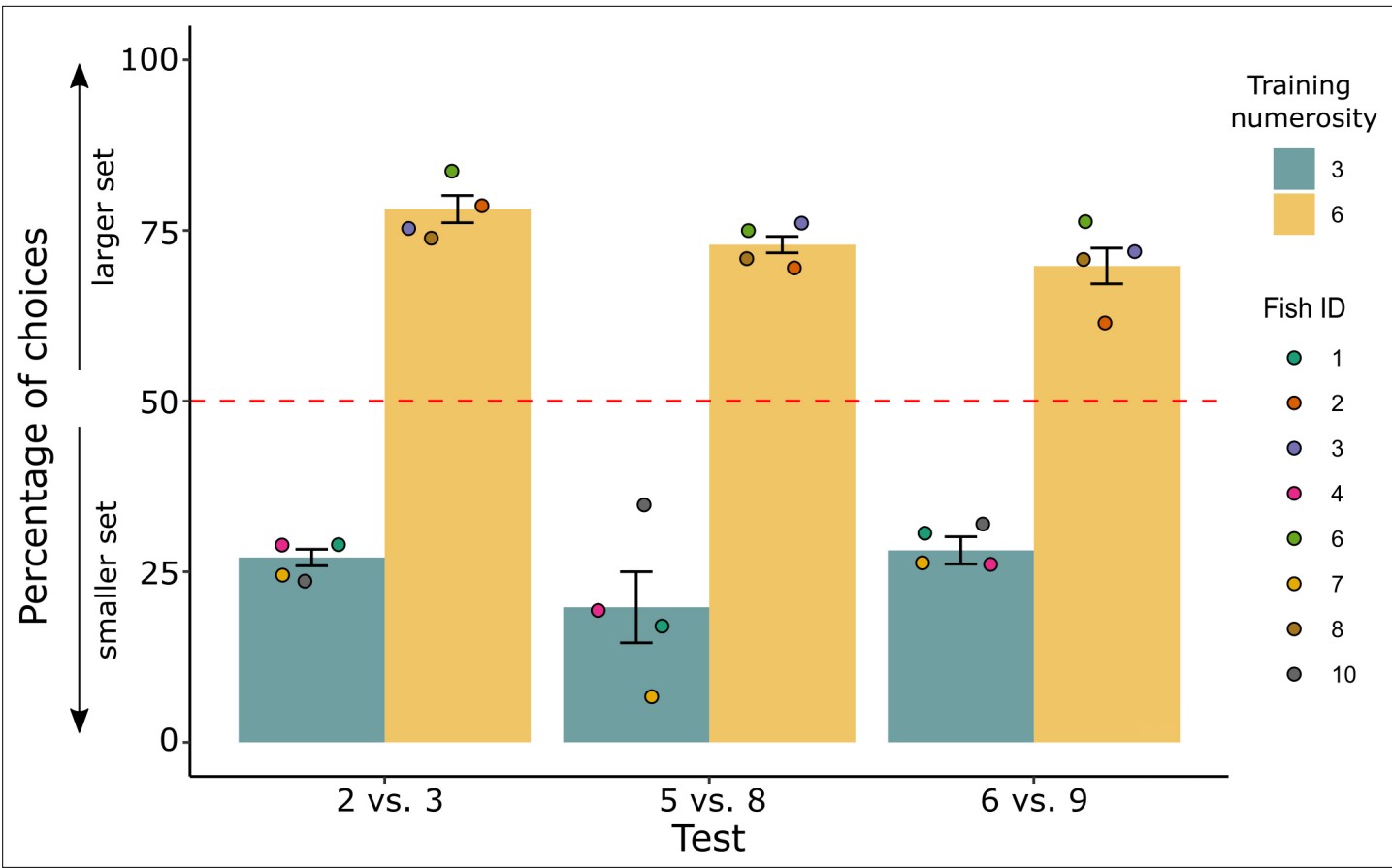

**Figure 3.** Percentage of choice for the larger/smaller set (mean ± standard error of the mean [SEM]) in the comparison tests for the two groups trained to select the smaller (3) or larger (6) set. Coloured dots represent the individual performance for each fish.

The online version of this article includes the following figure supplement(s) for figure 3:

**Figure supplement 1.** Individual performance data for each non-numerical control condition at test in Experiment 1.

Test 3: In the last test, fish behaviour was observed in a comparison involving novel numerosities never experienced during the training, that is, 5 vs 8. This allowed observing whether archerfish applied a relative representation (go for the 'smallest' or 'largest') or if the choice was at the chance level, since no absolute numerical information experienced at training was present here.

Results at tests for Experiment 1 are reported in *Figure 3* (see also *Figure 3—figure supplement 1* for individual fish' performance in each control condition). Choices for the relative numerosity were analyzed using a GLMM (see Methods section). Four fixed effects (type of training – 3 dots, 6 dots; type of test – 2 vs 3, 5 vs 8, and 6 vs 9; type of geometrical control – radius fixed, overall area controlled, overall perimeter controlled; type of spatial disposition control – inter-distance controlled, convex-hull controlled) and one random intercept effect (fish ID) were considered. Analysis of the random effect showed not to affect the model (random intercept variance of the best fit: $4 \times 10^{-14}$), and no significant differences were found between effects of groups, nor group interactions, suggesting to adopt the simplest model considering the choices with no contribution from any effect. Only a trend for the contribution of the type of geometrical control was observed, driven by a non-significant difference between the 'radius fixed' and 'overall area controlled' conditions (post hoc non-parametric tests adjusted with Tukey method: p = 0.063). Within this trend, every single condition was statistically significant by chance level in the direction of the relative choice (log odds ratio estimates: 0.90 ± 0.13 for radius; 1.47 ± 0.21 for overall area; 1.03 ± 0.19 for overall perimeter), as confirmed by exact binomial tests for each different group ('radius' estimate in natural units: 0.71, probability of success: 71.2%, p < 0.001, 95% CI: 65.6–76.3; 'overall area' estimate in natural units: 0.81, probability of success: 81.3%, p < 0.001, 95% CI: 79.3–87.3; 'overall perimeter' in natural units: 0.74, probability

of success: 73.6%, p < 0.001, 95% CI: 65.6–80.6. The corresponding Cohen's *g* (see Methods section) was, respectively, 0.21 – medium effect size – for 'radius', 0.31 – large effect size – for 'overall area', and 0.24 – medium effect size – for 'overall perimeter').

Considering the previous discussion, a binomial test shrinking all the data together was performed to investigate the final findings: fish showed an overall strong significant preference for the relative numerosity (probability of success: 74.3%, p < 0.001, 95% CI: 70.5–77.8, Cohen's *g* = 0.24 medium effect size).

The result obtained in Experiment 1 showed that archerfish, when trained to select one of two simultaneously displayed groups of dots with different numerosities (i.e. 3 vs 6 dots), use a relative numerical rule to perform novel numerical comparisons. These results confirm findings in other fish species such as angelfish (*Miletto Petrazzini et al., 2016*) and guppy (*Miletto Petrazzini et al., 2015*), but they are different from those obtained in bees which showed instead a preference for the absolute number (*Bortot et al., 2019*). An important difference between fish and bees studies is related to the numerical comparison used: respectively, large numbers (>4 elements) for fish and small numbers (≤4 elements) with bees. This might engage different systems (as reported in Introduction section), explaining the discrepancy. The training discrimination used in Experiment 1 involved two numbers (3 vs 6) that belong to the hypothesized 'small' and 'large' systems, respectively. This is different than in previous fish studies which employed only large numerosities; thus, it remains to be tested how fish would deal when trained with small numerosities only. In principle, the presence of a large number in the comparison in Experiment 1 may be enough to lead the archerfish to follow a relative rule. If trained with a numerical discrimination involving only small numbers, would the animals

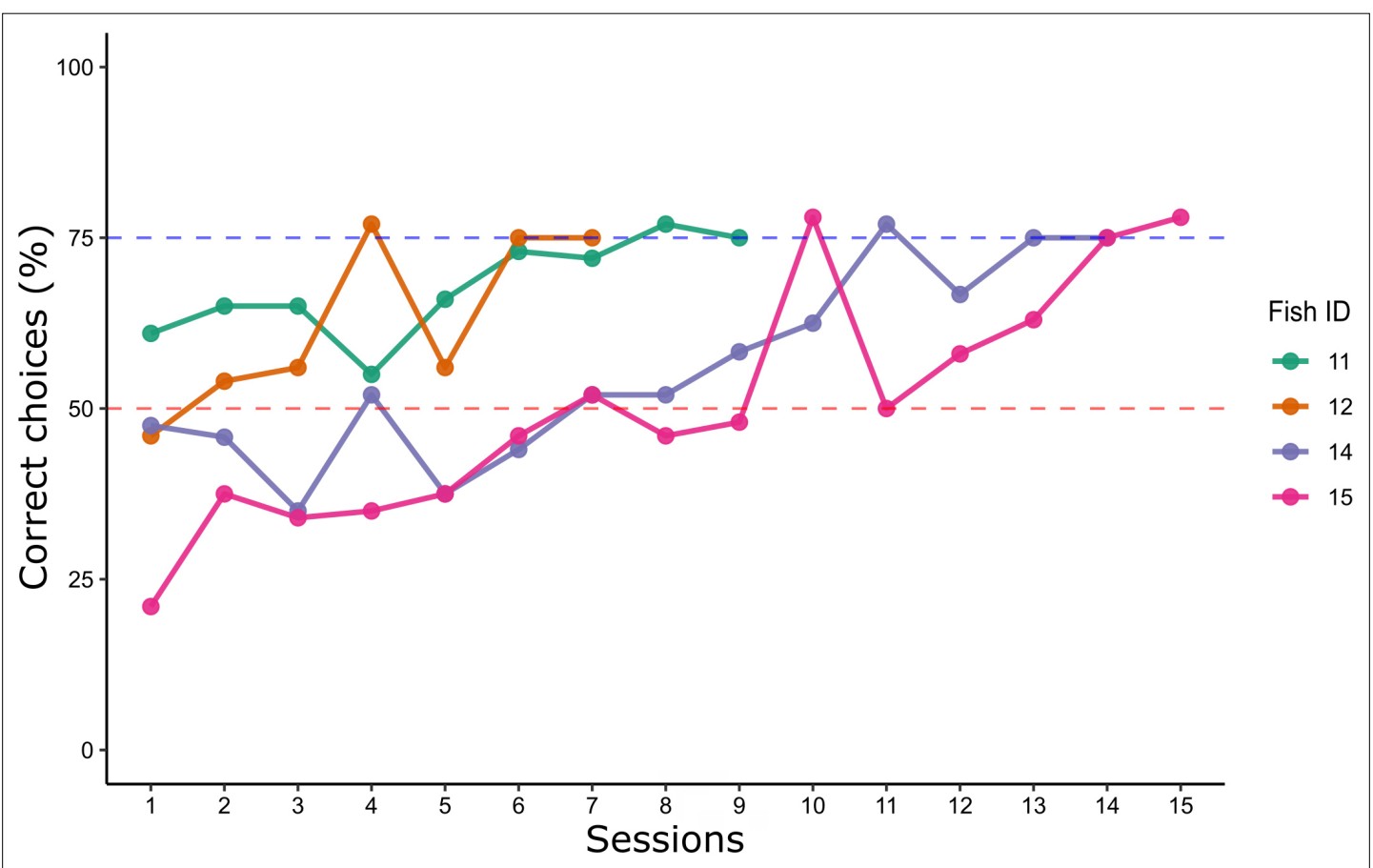

**Figure 4.** Learning curves of Experiment 2: the lines show the percentage of correct choices for each archerfish in a 2 vs 3 training. Learning criterion (blue dotted line) was reached after two consecutive sessions ≥75%. The red dotted line refers to chance level.

The online version of this article includes the following figure supplement(s) for figure 4:

**Figure supplement 1.** Performance data for each non-numerical control condition at training in Experiment 2.

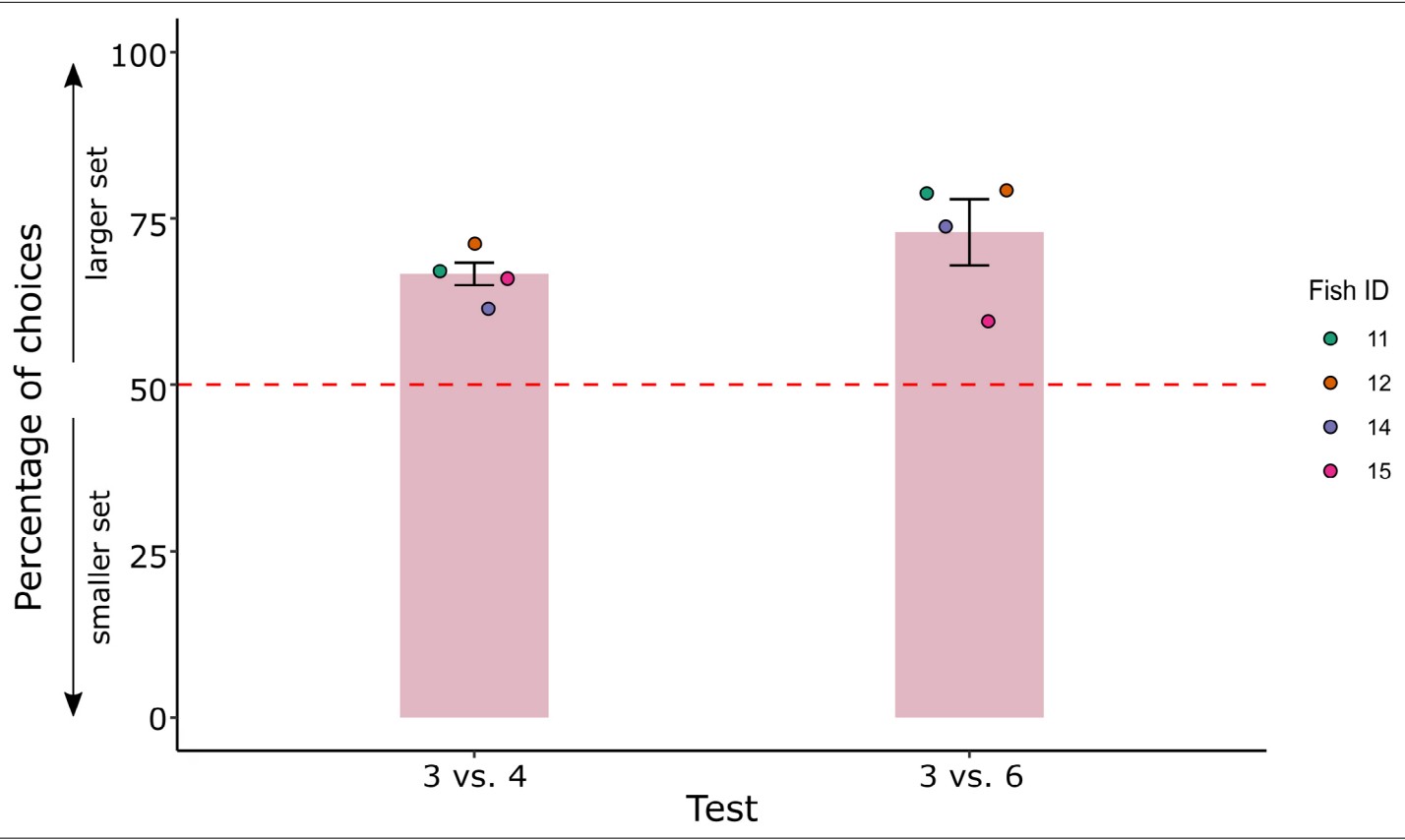

**Figure 5.** Percentage of choice for the larger set (mean ± standard error of the mean [SEM]) in the comparison test set of Experiment 2. Coloured dots represent the individual performance for each fish.

The online version of this article includes the following figure supplement(s) for figure 5:

**Figure supplement 1.** Individual performance data for each non-numerical control condition at test in Experiment 1.

still use a relative numerosity judgement or would they turn to absolute judgement? This was tested in Experiment 2.

## Experiment 2

Four subjects were trained to select the largest number in a 2 vs 3 comparison (i.e. the number 3). Fish judgement was then observed in two tests (i.e. 3 vs 4 and 3 vs 6) involving a comparison between the previously trained numerosity (3) and a novel numerosity (4 or 6).

All fish reached the learning criterion, showing an ability to discriminate between the two numbers (trials to criterion ± SEM = 506.5 ± 97.8; *Figure 4*).

Choices during the last two sessions (over criterion) were analyzed using a GLMM (see also *Figure 4—figure supplement 1* for individual fish' performance in each control condition). Two fixed effects (type of geometrical control – radius fixed, overall area controlled, overall perimeter controlled; type of spatial disposition control – inter-distance controlled, convex-hull controlled) and one random intercept effect (fish ID) were considered; the independent variable was the choice for the reinforced numerosity. Analysis of the random effect showed not to affect the model (zero variance for the random intercept of the best fit). No significant differences were found between effects of groups, nor group interactions, suggesting to adopt the simplest model (i.e. considering the fish' choices with no contribution from any effects). The best model final estimate (logarithmic odds ratio) was 1.12 ± 0.12, corresponding to 0.75 in natural units (fraction of choices) as confirmed by a binomial test (probability of success: 75.5%, p < 0.001, 95% CIs: 70.8–79.8, Cohen's *g* = 0.25 large effect size).

Results at test are reported in *Figure 5* (see *Figure 5—figure supplement 1* for individual fish' performance in each control condition). A GLMM model with three fixed effects (type of test – 3 vs 4,

3 vs 6; type of geometrical control – radius fixed, overall area controlled, overall perimeter controlled; type of spatial disposition control – inter-distance controlled, convex-hull controlled) and one random intercept effect (fish ID) showed no random effect of fish (zero variance of random intercept for the best fit), neither significant differences between groups and groups' interactions, suggesting to adopt the simplest model (i.e. considering the fish' choices with no contribution from any effects).

The best model final estimate (logarithmic odds ratio) was 0.84 ± 0.16, corresponding to 0.70 in natural units, as confirmed by a binomial test (probability of success: 69.8%, p < 0.001, 95% CIs: 62.7–76.2, Cohen's $g$ = 0.2 medium effect size).

In Experiment 2, archerfish showed to be able to discriminate between two different numerical groups of dots within the small numerical range. At test, fish preferred the novel numerosity to the familiar three items, in both 3 vs 4 and 3 vs 6 comparisons, confirming the use of a relative rather than absolute numerical rule. This evidence does not match with findings in bees, tested in the same numerical conditions, suggesting that the spontaneous engagement of relative/absolute rule to extract numerical information may be guided by different ecological pressures experienced by different species in their phylogenetic history. The spontaneous use of relative rules, when only numerical cues are available, suggests that among fish, it is more important to learn a general rule that is applicable to novel comparisons. It cannot be excluded that this strategy is adopted because it could be less demanding as to memory load than an absolute judgement strategy.

Considering the results of Experiments 1 and 2, it is apparent that archerfish can easily discriminate between small and large numerosity using the same rules, providing evidence in favour of a unique system underlying numerical discrimination as found in other fish species (*Stancher et al., 2013*; *Potrich et al., 2015*).

## Accuracy is not influenced by non-numerical magnitudes

The control of non-numerical magnitudes applied to our stimuli considers all the possible combinations associating the geometry and the spatial disposition of the numerical dots' array. Given that it is empirically impossible to control for all these physical factors at once, as a consequence, when some are balanced, others may be free to covary congruently with numerosity (i.e. as the numerosity increases, the non-numerical information increases as well). A previous study led by Leibovich-Raveh et al. in archerfish (*Leibovich-Raveh et al., 2021*) aimed to study whether the spontaneous choice for two numerically different groups of dots was influenced by how many physical variables were positively correlating with numerosity. By manipulating the geometry and spatial disposition of the elements, several stimuli with different congruity levels were created, ranging from 1 to 5 (i.e. from congruity level 1: only one physical variable was correlating positively with numerosity, to congruity level 5: all the five physical variables considered in the study were positively correlating with numerical information). The results showed that archerfish' choice for the largest or smallest numerosity in the study by Leibovich-Raveh and colleagues was indeed influenced by the non-numerical variables that positively correlated with numerosity. Using a similar approach, we checked whether the archerfish' performance accuracy in detecting one of the two numerical sets correlated with the number of non-numerical information that were positively covarying with numerosity.

**Table 1.** Schematic representation of the levels of congruity for each control condition (reported in the table rows) applied in the study; the columns represent the different variables that could covary with numerosity (C: congruent with number, IC: incongruent with number).

| Control condition | Overall area | Overall perimeter | Convex-hull (CH) | Inter-distance (ID) | Congruency level |
|---|---|---|---|---|---|
| Perimeter – CH | IC | = | = | C | 1a |
| Perimeter – ID | IC | = | C | = | 1b |
| Area – CH | = | C | = | C | 2a |
| Area – ID | = | C | C | = | 2b |
| Radius fixed – CH | C | C | = | C | 3a |
| Radius fixed – ID | C | C | C | = | 3b |

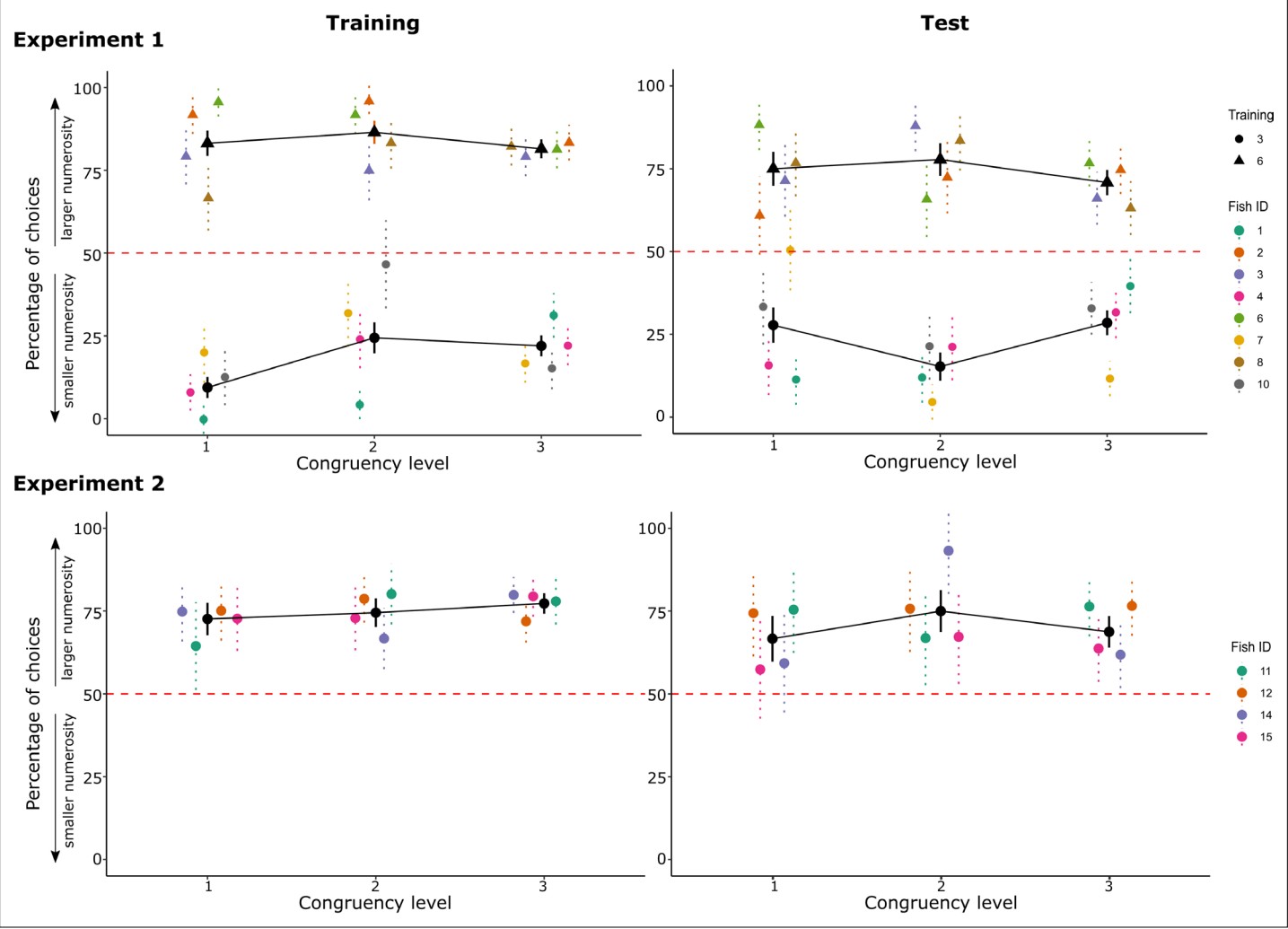

**Figure 6.** The graphs report the choices for the larger numerosity depending on the different levels of congruency: graphs are reported both for the training phase (mean data of the last two sessions when the criterion was reached) and the test phase, for both Experiments 1 and 2. In Experiment 1, data are grouped by training condition (circles for fish trained with three dots, triangles for fish trained with six dots). Coloured points represent single fish performance with standard error bars (i.e. data are mediated over trials with the same congruency level, per each fish), while black points represent the overall mean (i.e. data are mediated over all the trials with the same congruency level). Red dotted lines represent chance levels.

The online version of this article includes the following source data for figure 6:

**Source data 1.** Comparisons between different levels of congruity (p values of post hoc analysis with Tukey correction) and 95% confidence intervals for each level.

Analyzing the characteristics of our stimuli, the levels of congruity (i.e. the number of physical factors positively correlating with numerosity) ranged from 1 to 3, depending on the different stimuli configurations (see *Table 1*).

To evaluate the influence of different congruency levels on the fish' choice, an analysis was performed fitting a GLMM with the levels of congruency as fixed factor. The results (reported in *Figure 6*) proved that no correlation was present between the levels of congruity and the choices for the larger/smaller numerosity, both for Experiments 1 and 2 (differences between groups, given by a post hoc analysis with Tukey correction showed p values >0.05; see *Figure 6—source data 1*). This evidence suggests again that fish accuracy was not influenced by non-numerical variables, confirming that the relative rule adopted was mainly driven by the numerical cue. At first sight, this result could seem discordant from what established by Leibovich et al., who found that the increasing number of variables influenced archerfish performance. However, in our study, archerfish were trained to select the stimuli with specific numerosity, while in the Leibovich-Raveh's study, fish were observed in a

spontaneous choice task always rewarded, irrespective of the chosen stimulus. Taken together, these pieces of evidence might suggest that magnitude information matter and are particularly salient to archerfish, but they do not interfere when numerical rules are specifically engaged.

## Numerosity and spatial frequency

The stimuli used in our experiments were visual collections of black dots differing in numerosity. As described in Methods section, for each numerical comparison, the physical properties of each array were equalized for the geometry (radius, area, and perimeter) and spatial disposition (inter-distance and convex-hull; see *Figure 1*). Since we are dealing with images, each figure could also be described in terms of spatial frequency. Spatial frequency can be thought of as the number of repeating elements in a pattern per unit distance, and it is mathematically described by the Fourier transform theory. No control was applied to the spatial frequency of our stimuli. Thus, in order to check whether spatial frequency could influence archerfish choice, we calculated its variation across all different numerosities and control conditions (see Methods section). Within each numerical test comparison, different spatial frequencies were found (see *Figure 7*). The different constraints applied to the stimuli (control of the area, perimeter, or elements radius) showed to differently influence the spatial frequency between the two numerosities. In detail, when the elements' radius was fixed between the two numerical arrays, the total power of the spatial frequency was higher in the smaller group than in the larger one, while the opposite was found in the groups in which the overall perimeter was balanced (total power higher in the more numerous group). Interestingly, this trend was maintained in all the numerical comparisons used, irrespective of the number of elements to be compared.

To investigate the influence of spatial frequency in the numerical task, we analyzed whether a correlation between the performance accuracy and the spatial frequency was apparent, for all possible control configurations (see Methods section). Results are reported in *Figure 7*, showing no correlations between any comparison (test 2 vs 3: $r(4) = -0.17$, p = 0.83; test 3 vs 4: $r(4) = 0.15$, p = 0.77; test 3 vs 6: $r(4) = -0.35$, p = 0.50; test 5 vs 8: $r(4) = -0.08$, p = 0.88; test 6 vs 9: $r(4) = -0.42$, p = 0.41).

These data strongly suggest that the spatial frequency was not influencing archerfish performance in the numerical task.

## Discussion

Overall, our results showed that when trained to select a specific group of elements between two numerical arrays, archerfish spontaneously generalize at test to novel numerical comparison according to a relative numerical rule (select the largest/smallest) rather than an absolute numerical rule (select the specific number of items). These findings are in agreement with previous results from other fish species and humans (*Miletto Petrazzini et al., 2016*; *Miletto Petrazzini et al., 2015*), while differing with respect to bees (*Bortot et al., 2019*).

Interestingly, archerfish use a general relative judgement even when trained to discriminate between numerosities that belong to different systems, namely 'OTS' for small numerosities (≤4 elements) and 'ANS' for large numerosities (>4 elements; for a review see *Hyde, 2011*). In Experiment 1, archerfish were trained with a 3 vs 6 contrast and then observed in test conditions with a 2 vs 3, 6 vs 9, and 5 vs 8 comparison. In all these tests, archerfish showed to spontaneously use a general relative rule. In Experiment 2, subjects' performance was observed in a numerical discrimination involving only small numerosities (i.e. 2 vs 3) at training. Once again, at test, fish followed the relative rule, selecting the largest group in the test comparisons 3 vs 4 and 3 vs 6, thus ignoring the absolute number of elements (i.e. 3). Since previous findings in vertebrate species showed the use of relative numerosity judgements only in comparative assessments for large numerosities (>4) (*Miletto Petrazzini et al., 2016*; *Miletto Petrazzini et al., 2015*); here, we provide evidence that the same rule is engaged even when comparisons involve both small and large numerosities together (Experiment 1) as well as small numerosities only (Experiment 2). Moreover, the latter condition offers us, for the first time, a direct comparison with evidence in invertebrates (tested only with small numbers) (*Bortot et al., 2019*).

Taken together, our results support the hypothesis of a unique system for representing numerosities in archerfish, working both for small and large numbers, obeying the ANS. Evidence from other fish species supports this claim (*Stancher et al., 2013*; *Potrich et al., 2015*).

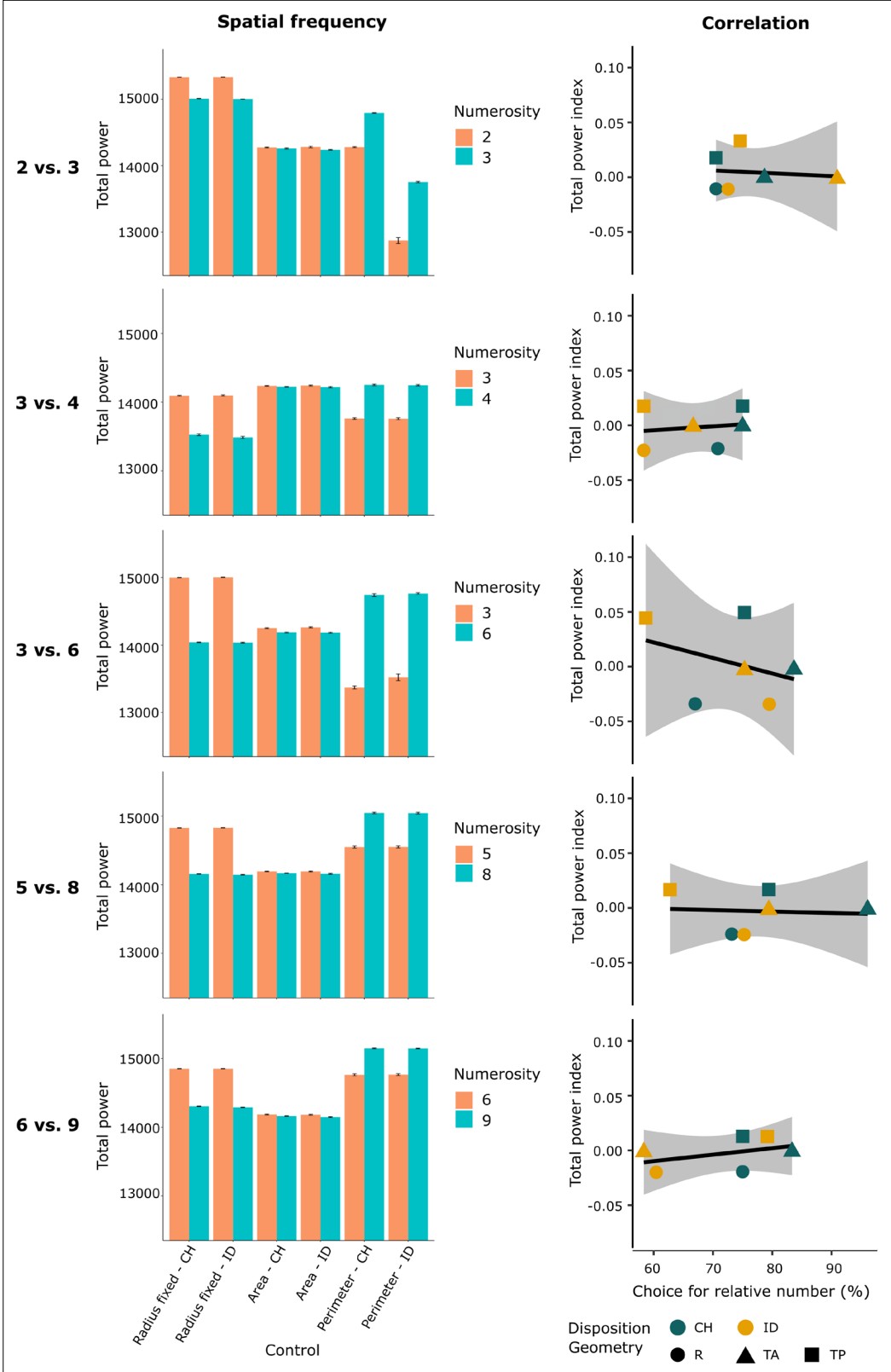

**Figure 7.** The histograms (on the left) show the spatial frequency (total power) for each numerical comparison among the different control groups (non-numerical variables control). The different constraints applied to the stimuli (control of the area, perimeter, or elements radius) showed to influence the spatial frequency between the two compared numerosities. The regression lines (on the right) show the correlation between fish' performance

*Figure 7 continued on next page*

*Figure 7 continued*

accuracy (choice for the relative numerosity) and the spatial frequency (total power index between the two total power values), for all numerical comparisons. The coloured shapes (dots, triangles and squares) correspond to each specific control condition.

The reason for which archerfish primarily rely on the relative information of numerical groups may have ecological reasons, being more adaptive in a natural environment that constantly require numerical/quantity judgement. Selecting the largest social group of companions or the largest food patch are easy rules that can be more efficient than using an absolute rule. Moreover, the use of relative information may be less cognitively expensive (in terms of memory load) than the absolute one, since it does not require storing the information about the precise number of elements: the discrimination could work on a simple relative comparison between magnitudes, guided by the ratio between the two. Nevertheless, the engagement of relative rules requires a good level of abstraction and the creation of a general rule to be applied to (*Pepperberg and Brezinsky, 1991*).

In fish, the use of an absolute rule may not be as convenient as the relative one, given that in most ecological contests there is no specific optimal amount of food, partners or companions. However, this seems not to be the case for species such as bees, which showed instead a spontaneous use of absolute numerical information, suggesting that this rule may be more informative and useful in their ecological environment. Similar evidence has been found in spiders, that, in a natural predatory strategy context, settle their attack based on the specific number of conspecifics at the nest (*Nelson and Jackson, 2012*).

Note, however, that the spontaneous use of a relative or absolute rule does not imply that animals are unable to use both. Vertebrates can be trained to learn a specific number of items in a set if forced to do it (*Cantlon and Brannon, 2007*; *Miletto Petrazzini et al., 2015*; *Pepperberg, 1994*; *Smirnova et al., 2000*). Similarly, bees can be trained to the numerical concepts of 'greater than' or 'smaller than' (*Howard et al., 2018*). The spontaneous engagement of one of the two criteria is therefore justified probably by a combination of natural constraints and/or less cognitive demand motivations that better fit for the individuals' fitness in their particular niches of adaptation. Note that, despite our study provides direct evidence of archerfish's ability to learn abstract numerical rules, this does not directly imply that number is spontaneously used in an ecological environment. Here, fish were guided to use numerical cues since it was the only variable systematically reinforced. In more ecological settings, continuous variables highly correlate with numerosity, with a consequent difficulty to understand on which type of information archerfish spontaneously primarily rely on and whether this could push fish to use absolute or relative magnitudes.

Lastly, with respect to the main question of our paper, the results showed that archerfish are capable of abstract numerical discrimination, not influenced by other continuous physical variables. We tested archerfish with numerical arrays well controlled for all the possible non-numerical variables (e.g. total area, perimeter, inter-distance, density, and convex-hull). Note that previous studies in fish that have attempted to control for non-numerical variables during the learning process were mainly focused on the overall elements' area, the elements' density, and convex-hull (see e.g. *Agrillo et al., 2012*; *Bisazza et al., 2014*; *DeLong et al., 2017*). In our study, all the geometrical constraints were controlled. These non-numerical cues did not correlate with the animals' performance accuracy (as demonstrated by the statistical analyses), ensuring that the discrimination made by the animals was based on purely numerical information.

Moreover, for the first time in fish, we focused our attention on analyzing spatial frequency of the stimuli used, showing that this variable does not influence archerfish performance. The total power of the spatial frequency has been described in the literature to positively increase with numerosity (*MaBouDi et al., 2021*); however, in our stimuli, the different geometrical constraints showed that it could be reversed as well. Moreover, elements' area and perimeter seem to play a crucial role in the distribution of the spatial frequencies' energy with respect to the elements disposition (inter-distance and sparsity). All our analyses suggested that the amplitude component of the spatial frequency was not influencing archerfish numerical evaluation during our experiments, providing useful information for further detailed investigations on the contribution of such variable in fish cognition.

Note, however, that in all studies carried out so far (including our own analysis), the focus was on the amplitude of the spatial frequency as the main component, which provides information on the

alternation rate of different elements in the image. It is likely that a more specific role on computation of numerosity is played by the spatial frequency phase component (related to elements' spatial coherence and distribution) which directly relates to figure-ground segregation and unity formation.

In conclusion, our results provide clear evidence that under conditions of strict control of continuous physical variables, archerfish can encode an abstract concept of number to support relative numerical judgement for both small and large numerosities.

## Materials and methods
### Subjects and rearing conditions

Sixteen adult archerfish, *T. jaculatrix* (fish size ranged between 8 and 10 cm in length) were provided by a local commercial supplier ('Acquario G di Segatta Stefano', Trento, Italy). A group of fish (*N* = 8) took part in Experiment 1, while a second group (*N* = 4) took part in Experiment 2. Fish were randomly assigned to the two training conditions. The other four animals were excluded because they did not show any consistent motivation in hitting the screen, failing to get through the different steps of the pre-training phase (see 'General procedure' paragraph). All fish were housed in large aquariums (100 × 40 × 40 cm) in groups of 10 individuals. Prior to the experiment, each archerfish was moved into individual aquaria (40 × 30 × 50 cm) filled with freshwater maintained at 25°C and enriched with gravel and a shelter. Water quality was kept by suitable filters (Sera fil 60). The system was illuminated under a 10:14 light/dark cycle (Sylvania luxline plus F36W/840 cool white). Fish were fed with food pellets (Hikari cichlid gold baby pellet).

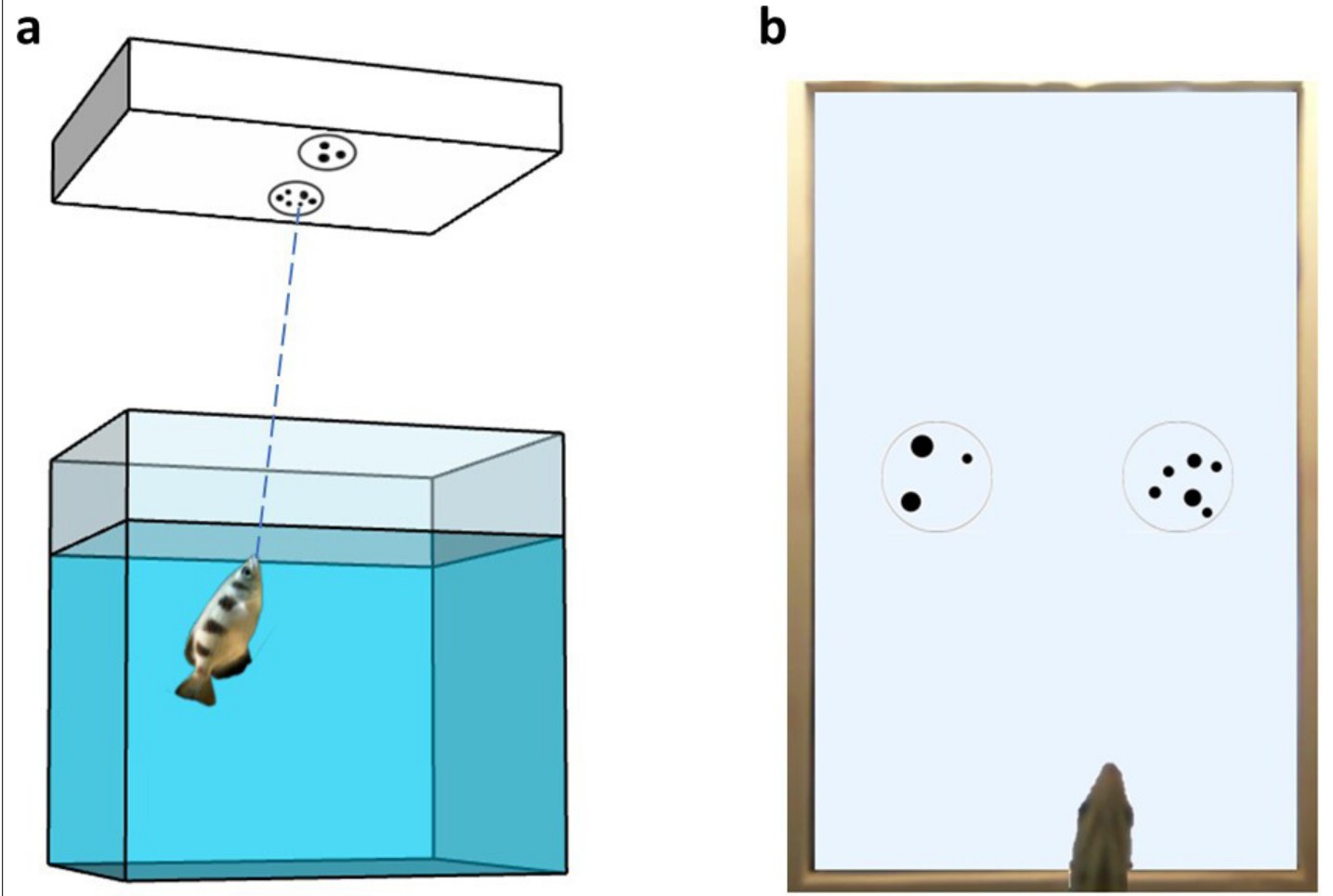

**Figure 8.** Experimental setup.
(**a**) Schematic representation of the experimental apparatus. (**b**) Bottom view of the tank from the camera placed below the tank's pavement.

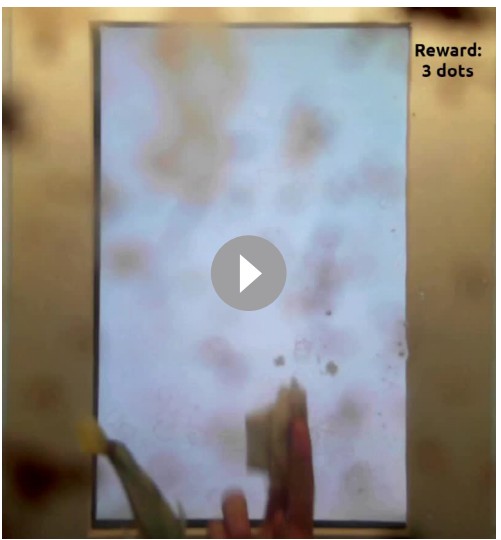

**Video 1.** Video example of a subject performing a 2 vs 3 discrimination during the training phase (Experiment 2; three dots stimulus is rewarded). The video camera records a bottom view of the fish and the screen.
https://elifesciences.org/articles/74057/figures#video1

## Apparatus

Both the apparatus and the training method were set up based on previous studies conducted with archerfish on visual discrimination tasks (i.e. *Karoubi et al., 2017*; *Newport et al., 2013*; *Ben-Tov et al., 2015*). Each experimental tank consisted of a rectangular aquarium with a monitor screen located above it (20″, DELL 2009Wt), held at 30 cm from the water level (*Figure 8a*). Each tank was surrounded by white opaque panels to ensure that the fish was not distracted by external cues. Each tank was raised 8 cm off the table thanks to lateral supports, allowing the positioning of a video camera under the centre of the pavement's tank to record a bottom view of the fish and the screen (see *Videos 1 and 2* examples in the supplement materials).

## Stimuli

The stimuli presented in the training phase consisted of groups of black dots confined into a black outline circle (6 cm diameter). The dots size was ranging between 3 and 12 mm, and the visual angle was in the range 0.43° and 1.72°, which has been proven to be well perceived by archerfish (*Ben-Simon et al., 2012*). In every trial, a couple of stimuli was simultaneously presented in the centre of the screen (horizontally aligned to the shortest monitor's side, see *Figure 8b*). All the stimuli were created using the software GeNEsIS (*Zanon et al., 2021*), a Matlab program that allows to create numerical collections of stimuli controlled for several non-numerical magnitudes. Given that it is mathematically impossible to balance all the non-numerical magnitudes simultaneously in two different numerical groups (e.g. when the convex-hull of the stimuli increases, the density decreases and vice versa; similarly, when the overall area of two sets of elements with different numerousness is balanced, their overall perimeter differ, etc.), different sets of stimuli were created for each numerosity, controlling for some visual physical property; all the possible properties were covered across the different sets during a session (see *Figure 1* for a view of all the combinations applied in a session and *Table 1* to see the variables balanced and not balanced in each condition). Doing so, even some physical variables were not controlled in one specific condition (e.g. when the overall area of the two sets was balanced, their overall perimeter differ; when the convex-hull of the stimuli increases, the density decreases, and vice versa), the use of different randomized control conditions allowed us not to make any of the physical variables systematically reliable and rewarded. Pictures from each set were randomly presented, making the numerical information the only reliable cue to differentiate the two stimuli across all the various trials.

## General procedure
### Pre-training phase
Before starting the experiment, fish underwent a pre-training phase in which they were gradually habituated to spit (hit with a jet of water) at the training stimulus on the screen. This was accomplished throughout a shaping procedure to facilitate the task. The silhouette of an insect was initially presented, inducing the fish reaction to

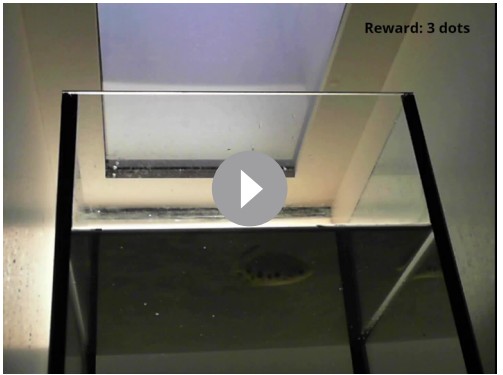

**Video 2.** Video example of a subject performing a 3 vs 6 discrimination during the training phase (Experiment 1; three dots stimulus is rewarded). The video camera records a lateral view of the fish and the screen.
https://elifesciences.org/articles/74057/figures#video2

spit at the prey; once hit, fish were rewarded with a food pellet. The insect was gradually replaced by a black dot and finally with the effective training stimulus. Once the fish accomplished all these stages, the training phase was initiated. As mentioned above, four animals did not achieve this phase, due to the fact that animals were rarely interested in hitting at the screen (mostly motivated when the insect silhouette was presented but not with images depicting a dot). After 10 consecutive sessions with the aforementioned passive behavior, animals were excluded from the study.

### Training phase

Fish were trained to spit at the correct target presented on the monitor above the tank. The stimuli to discriminate consisted of two groups of dots with different numerosity. Every trial started with the appearance of a blinking black square (1.6 cm, three blinks of 100 ms) at the centre of the screen to catch the fish's attention towards the screen. Then, the two training stimuli were displayed one next to the other (distance 7 cm) on the two sides of the monitor. Only one of the two numerosities was rewarded with a food pellet when hit, while the choice for the incorrect stimulus caused the stop of the trial, which in every case, in absence of choice, was stopped after 5 min. At the end of each trial, the screen was cleaned from the water drops and a new trial started.

In the first training session only, a corrective method was applied: the stimuli remained on the screen until the subject selected the correct target, even if the incorrect stimulus was hit, allowing the fish to correct its choice.

Fish were trained with daily sessions of 48 trials, in which continuous physical variables were controlled and changed according to the scheme reported in *Figure 1*, and the position of the target stimulus on the screen (right–left) was randomized. In detail, on the total of 48 randomized trials, the control conditions were organized as follows: 12 trials for each condition 'radius fixed – inter-distance' and 'radius fixed – convex-hull'; 6 trials for each condition 'area – inter-distance', 'area – convex-hull', 'perimeter – inter-distance', and 'perimeter – convex-hull'. In this way, half of the trials had the same individual size for each dot (radius fixed conditions), allowing us to control that the fish would not pay attention to the smaller or larger elements in size. Note that this is a salient cue, quite always appearing when the area and perimeter are balanced (since more/less numerous sets will naturally present the smallest/largest dots' extension, respectively). With our randomization approach, if such a cue would be relevant to fish, it should result in a final performance at chance level in the 'radius fixed' conditions, but not in the others ('area' and 'perimeter'). As a consequence, a significant difference among control conditions would emerge. Similar considerations can be made for the other controlled variables.

Fish generally responded 70–100% of the trials. The learning phase was considered completed when the fish reached a learning criterion of at least 75% of correct choices for two consecutive days (binomial test: p < 0.01), allowing the fish to take part in the test phase.

### Test phase

Generally, each test condition consisted of the presentation of a couple of stimuli with a novel numerical comparison, aiming to see if the target numerosity learned in the training phase was represented as a relative or an absolute numerical information. Each test was composed of 24 probe trials not rewarded, divided into three testing days of 8 randomized trials containing all the control conditions (2 trials for each condition 'radius fixed – inter-distance' and 'radius fixed – convex-hull'; one trial for each condition 'area – inter-distance', 'area – convex-hull', 'perimeter – inter-distance', and 'perimeter – convex-hull'). In each test session, the eight test trials were shuffled and interspersed with rewarded recall training trials (32 recall in total) to maintain the fish motivation high during the whole test duration. The order of the tests was randomized among the fish to exclude that the performance could be influenced by their order. At the end of each test, the fish underwent a complete daily session of retraining to further exclude potential interference among the tests.

## Statistical analyses and data analysis

Data were analyzed using R software (R-4.1.0). In Experiment 1, an independent *t*-test was used to compare the number of trials to reach the criterion between the two groups at training. For the last two training trials (over criterion) and at test, choices for the relative numerosity were analyzed using a GLMM fit by maximum likelihood (Laplace Approximation), binomial GLMM with a logit link.

The best model was selected after a back elimination procedure, removing interactions and factors iteratively, and comparing the different models based on AIC and BIC information criteria. A final binomial test was used to compare the distribution of the choices for the relative and absolute numerosities when no factors were significantly contributing to the results. Log odds ratios from the best fits were reported as GLMM estimates with their errors and converted in natural scales to give a more straightforward interpretation of the effect size (which for our binomial distributions corresponds to the sample proportion). 95% CIs were also reported for a cleaner interpretation of the final results. Moreover, following *Cohen, 2013*, the chance proportion of 0.5 was subtracted to our binomial sample proportion to obtain a final Cohen's g effect size (interpretable as: <0.05 negligible, 0.1–0.15 small, 0.2–0.25 medium, >0.25 large). To obtain an estimate of the spatial frequency, we adopted an approach already performed in other studies (*MaBouDi et al., 2021*; *Adriano et al., 2021*; *Felisatti et al., 2020*): the fast Fourier transform of our images was calculated, a radial average of the signal amplitude in the frequency domain was performed, and lastly, all the frequency contributions of its power spectrum were summed up. In this way, a value related to the total energy of each frequency component inside a given image is obtained.

To investigate the influence of spatial frequency in the numerical task, we analyzed whether a correlation between the performance accuracy (choice for the relative numerosity) and the spatial frequency (normalized total power difference between the two compared numerosities) was apparent for all possible control configurations. To compare two numerosities we reported a normalized difference (total power index) between the two total power values (difference between the total power of the biggest numerosity and the smallest, divided by their sum). All the frequency calculations were performed with a custom script in Matlab (https://github.com/MirkoZanon/GeNEsIS; *Zanon, 2021* copy archived at swh:1:rev:e2c1e1a12e033ffe2aa623b2ebb3f97fb5ea26a8), while the statistical comparisons were calculated in R. For each of them, a Pearson's correlation coefficient was calculated comparing the choice for the relative numerosity and the normalized difference between numerosities (as explained above).

## Acknowledgements

Funding: This project has received funding from the European Research Council (ERC) under the European Union's Horizon 2020 research and innovation programme (grant agreement no. 833504 SPANUMBRA) and Prin 2017 – Title: 'Number-space association: a comparative, developmental and neurobiological approach' (codice progetto 2017PSRHPZ) awarded to GV.

## Additional information

### Funding

| Funder | Grant reference number | Author |
| --- | --- | --- |
| H2020 European Research Council | 833504 SPANUMBRA | Giorgio Vallortigara |
| Progetti di Rilevante Interesse Nazionale | 2017PSRHPZ | Giorgio Vallortigara |

The funders had no role in study design, data collection, and interpretation, or the decision to submit the work for publication.

### Author contributions

Davide Potrich, Conceptualization, Data curation, Formal analysis, Investigation, Methodology, Project administration, Validation, Writing - original draft, Writing - review and editing; Mirko Zanon, Data curation, Formal analysis, Methodology, Software, Writing - review and editing; Giorgio Vallortigara, Conceptualization, Data curation, Funding acquisition, Project administration, Resources, Supervision, Writing - original draft, Writing - review and editing

### Author ORCIDs

Davide Potrich http://orcid.org/0000-0003-0928-628X

Mirko Zanon  http://orcid.org/0000-0003-4062-1496
Giorgio Vallortigara  http://orcid.org/0000-0001-8192-9062

## Ethics

The present research was carried out at the Animal Cognition and Neuroscience Laboratory (ACN Lab) of the CIMeC (Center for Mind/Brain Sciences), at the University of Trento (Italy). All husbandry and experimental procedures complied with European Legislation for the Protection of Animals used for Scientific Purposes (Directive 2010/63/EU) and were approved by the Scientific Committee on Animal Health and Animal Welfare (Organismo Preposto al Benessere Animale, OPBA) of the University of Trento and by the Italian Ministry of Health (Protocol no. 932/2020-PR).

## Decision letter and Author response

Decision letter https://doi.org/10.7554/eLife.74057.sa1
Author response https://doi.org/10.7554/eLife.74057.sa2

## Additional files

### Supplementary files
• Transparent reporting form

### Data availability

All data generated or analysed during this study have been deposited in Dryad.

The following dataset was generated:

| Author(s) | Year | Dataset title | Dataset URL | Database and Identifier |
|---|---|---|---|---|
| Potrich D, Zanon M, Vallortigara G | 2021 | Numerical discrimination of sets of elements by Archerfish | https://doi.org/10.5061/dryad.4f4qrfjdg | Dryad Digital Repository, 10.5061/dryad.4f4qrfjdg |

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
