## [Decision Letter]

**Decision letter after peer review:**

Thank you for submitting your work entitled "Archerfish number discrimination" for consideration by *eLife*. Your article has been reviewed by 3 peer reviewers and the evaluation has been overseen by Floris de Lange (Senior Editor) and a Reviewing Editor. The reviewers have opted to remain anonymous.

The reviewers were mainly positive about the research and the paper. In reviews and discussion, a number of serious issues were raised which the authors should address in their revised manuscript. Below are the most important points, which are considered essential revisions. The individual reviewer comments with additional detail are appended to this list for the authors' reference.

1. Sparsity of description. The description of the methods and data, as well as the level of analysis in its current form are insufficient to support the conclusions of the study in its present form. For instance, the number of trials per condition per fish needs to be reported, as well as the performance data for individual fish. Please see the specific suggestions by the reviewers.

2. Non-numerical parameters. While it is acknowledged that skillful attempts have been made to control for non-numerical parameters, the reviewers are still critical about this issue. It will be essential to address this concern specifically in a revised version of the paper. For instance, it will be necessary to present the subjects' behavior for the different control conditions and to show that different non-numerical factors were not or barely influencing performance. The individual reviews mention several additional analyses that need to be performed and more data to be shown.

3. Ethological preference. It is questionable whether the data support the claim that fish have a spontaneous preference for relative versus absolute magnitude. If the authors have data on the learning part and can analyze them, this might be a way to address this issue. If this claim can not be supported unambiguously, I suggest to delete the last sentence of the abstract and to mellow the discussion on this aspect.

4. Novelty. It is not entirely clear to what extent the current study surpasses previous number studies with trained fish. It will be important to clearly state the conceptual advancement that has been made relative to other fish training studies, most notably by the group of Bisazza and Agrillo (e.g. Agrillo et al., (2012) A new training procedure for studying discrimination learning in fishes; Bisazza et al., (2014) Extensive training extends numerical abilities of guppies.). A most relevant study that is not even cited is the one by DeLong et al., (2017), who trained goldfish to discriminate arrays of different numbers of dots by also applying controls for three non-numerical factors. Maybe the advancement is related to better controlled stimuli in combination with the demonstrated transfer of more/less rules to novel numerical quantity. In any respect, it needs to be clarified precisely why this fish training study is novel enough to warrant publication.

*Reviewer #1:*

In the study, archerfish were trained to choose between two stimuli that each contained a certain number of dots. After a learning criterion was achieved a test phase was introduced in which it was demonstrated that fish continued to choose targets according to the relative number of dots that was rewarded at training (either the larger or smaller set). The stimuli were designed such that they controlled for continuous physical variables to ensure that those variables could not account for the preferences of the fish. The results demonstrate that fish are able to learn an abstract relative numerical rule. Although the results are convincing, it is still not clear whether those continues variables that were not controlled in any given condition of geometrical control may have influenced performance as well (as it is impossible to control for all continues variables simultaneously). In addition, during the learning phase, the only predictive variable for reward was numerosity, but on different trials different continuous physical variables were still correlated with numerosity. Although numerosity was the only consistent predictive variable, rendering it much more salient for learning, it is still possible that other continuous variables still played a role in the learning process. For instance, when radius was controlled for, a summation of the area and perimeter might still impact performance. This might have biased learning towards the use of relative numerical rule and not precise numerical representations. As a result, it seems that the authors' claim for "spontaneous use of abstract relative numerical information in archerfish" might be somewhat misleading.

1) I was wondering if a similar approach to that employed by Leibovich-Raveh et al., could also be used in the current study? Can the authors examine the influence of the other continuous variables that were not controlled for in each test condition on the fish performance?

2) If my first comment can be addressed, it should still be emphasized that archerfish can learn to use abstract numerical rules, but that this conclusion might be restricted to conditions in which this is the only variable that is systematically reinforced, and that this might not represent the way archerfish process numerosity in more ecological settings, in which continues variables are highly correlated with numerosity.

*Reviewer #2:*

The authors have designed an elegant set of experiments to test the non-symbolic enumeration abilities of archerfish. These fish present a unique spitting behaviour to indicate their choice, so they are quite an ideal fish model to train and test in these types of choice behaviour. The experiments in this paper describe that archerfish generalise rules from a learned set of stimuli to a novel set. In each trial, fish are presented with two choices, each a display containing a number of black dots, of which the fish can consider and choose one. Once they learn to choose either the smaller of the two sets (containing fewer dots) or the larger, the authors use interspersed probe trials to test the basis on which the fish make their choice. The authors conclude that archerfish use the relative magnitude rule they learn during training to apply to new set of numbers.

The strengths of the paper are in the complete set of stimuli used in these experiments and the choice of number sets to address the question whether fish use relative versus absolute magnitude. The authors have created a rich set of stimuli including many controls for visual features that co-vary with the number of dots, like total visual area, perimeter and the convex hull. The authors additionally do a spatial frequency analysis of the stimuli used, which is novel and adds another visual control for the stimuli.

Unfortunately, the main question this manuscript addresses is familiar territory. Similar results have been reported in other species of fish, including studies by the authors. A common finding across these studies makes us expect that fish use relative magnitude rather than absolute magnitude. The authors discuss this in the manuscript as well.

Another weakness in the paper is that the methods and results are missing some crucial details, not including precise numbers of trials and the effect sizes of the tests, for example.

This sparsity of description makes it hard to judge whether the conclusions that the authors draw are well supported.

This dataset is valuable to the community as part of a growing set of studies investigating the numerical abilities of various species and relating these to their environment and ethology.

Major concerns:

1. The manuscript is missing a justification for the numbers used, especially when the authors compare the numbers used to those used in other fish studies or bee studies and state that '3' belongs to a small number, perhaps can be processed by the object tracking system rather than the approximate number system (for larger numbers).

2. To support the claims that fish have an ethological preference for relative versus absolute magnitude, the authors would need to show additional data, perhaps at the learning stage, of trials to criterion to learn absolute magnitude.

3. The description of the methods and results are lacking in a level of detail necessary to evaluate the manuscript, some of which I list below:

4. The average number of trials that each fish performed are never mentioned, but since there are only 24 test trials, 8 presented per day for 3 testing days, some fish might not see all the stimuli. Across 3 tests in Experiment 1, fish would have to see 6 control images for each number set. I'm not sure how that is accomplished.

5. The bar plots in Figures 2 and 3 should plot each individual fish and describe what the error bars describe.

6. Effect sizes need to be reported alongside the p-values, for the GLMMs as well as the other analyses.

7. Figure 4 contains no description of what the individual dots on the right side correspond to. There are 6 dots in each panel.

8. Were the control sets randomized or were they changed by session. Figure 1 gives the impression that they are deployed in individual test sessions.

9. Were the fish assigned to num 3 and num 6 training randomly? Or was this based on some shaping criteria? Additionally, please describe the criteria used to exclude the 4 fish. How many no-choice trials?

*Reviewer #3:*

The manuscript is devoted to studying archerfish number discrimination. The authors rely on their own generated computer code to generate stimuli for number discrimination task. They control some visual variables that are usually correlated with numbers of objects. The study is interesting and contributes to the growing evidence on numerical capabilities in branches of evolution far from mammals.

1. The evidence that non-numerical variables cannot influence the fish decision is light. The evidence relies on a computer code presented elsewhere. While I am fully aware that controlled everything is hard (or even impossible), I suggest that the author will make efforts to convince the reader that some visual variables were controlled and explain which variables were not controlled.

2. The data is reported on the entire population level. I think that single fish data is critical to assess the results. The success rate reported might be due to single fish which maintains high performance.

3. I assume that not all fish succeeded in the task and the reported fish are only fish which managed to achieve a criterion. A report on this part of the study will be useful.

4. The authors have data on the learning part, I suggest presenting and analyzing it.

5. Method training phase: the experimental procedure is identical to Ben-Tov 2015, this should be mentioned.

6. Given lines 222-225: what is new in the current study?

7. Discussion: As the author acknowledged, Leibovich-Raveh et al., showed that when archerfish make magnitude-related decisions, their choice is influenced by the non-numerical variables that positively correlate with numerosity. A clear statement why the current study is different is needed (see comment 1 above).

---

## [Author Response]

The reviewers were mainly positive about the research and the paper. In reviews and discussion, a number of serious issues were raised which the authors should address in their revised manuscript. Below are the most important points, which are considered essential revisions. The individual reviewer comments with additional detail are appended to this list for the authors' reference.1. Sparsity of description. The description of the methods and data, as well as the level of analysis in its current form are insufficient to support the conclusions of the study in its present form. For instance, the number of trials per condition per fish needs to be reported, as well as the performance data for individual fish. Please see the specific suggestions by the reviewers.

We provided all the missing information in the revised version. In the method section, we added detailed information regarding the number of trials per condition, the criterion of exclusion of subjects from the experiment and a description of how the physical variables were varying concerning each control condition.

The result section has been revised. New graphs have been added, showing the performance of each individual fish both at training and test. In addition, performance data for individual fish are reported in the Supplementary Figures.

2. Non-numerical parameters. While it is acknowledged that skillful attempts have been made to control for non-numerical parameters, the reviewers are still critical about this issue. It will be essential to address this concern specifically in a revised version of the paper. For instance, it will be necessary to present the subjects' behavior for the different control conditions and to show that different non-numerical factors were not or barely influencing performance. The individual reviews mention several additional analyses that need to be performed and more data to be shown.

In order to check whether non-numerical factors were influencing the fish accuracy, in the revised version, we analyzed archerfish performance in relation to each different constrained condition, both for the learning phase (considering the sessions when the learning criterion was reached) and the test conditions. Moreover, data on individual performance for each control condition are reported now in the Supplementary Materials. Our findings are strongly supported by the statistical analysis, which suggests no difference in performance among the various constraints.

Lastly, as suggested by the Reviewers, we deeply investigated whether a correlation between magnitude information and numerosity was present (using a similar approach to the one used by Leibovich-Raveh and colleagues). This additional analysis was added to the main text of the revised manuscript.

3. Ethological preference. It is questionable whether the data support the claim that fish have a spontaneous preference for relative versus absolute magnitude. If the authors have data on the learning part and can analyze them, this might be a way to address this issue. If this claim can not be supported unambiguously, I suggest to delete the last sentence of the abstract and to mellow the discussion on this aspect.

We clarified that the term “spontaneous choice” was related to the specific experimental condition in which fish were trained, namely only when numerical information was available: thus, the argument is that when given a possibility for option, fish spontaneously rely on relative rather than absolute magnitude. We hope to have clarified this issue in the revised version. We also specified that in more ecological settings, continuous variables highly correlate with numerosity, with a consequent difficulty to understand on which type of information archerfish spontaneously primarily rely on and whether this could favour fish to use absolute or relative magnitudes. We also added analysis of the learning part, as requested by the Editors and Reviewers.

4. Novelty. It is not entirely clear to what extent the current study surpasses previous number studies with trained fish. It will be important to clearly state the conceptual advancement that has been made relative to other fish training studies, most notably by the group of Bisazza and Agrillo (e.g. Agrillo et al., (2012) A new training procedure for studying discrimination learning in fishes; Bisazza et al., (2014) Extensive training extends numerical abilities of guppies.). A most relevant study that is not even cited is the one by DeLong et al., (2017), who trained goldfish to discriminate arrays of different numbers of dots by also applying controls for three non-numerical factors. Maybe the advancement is related to better controlled stimuli in combination with the demonstrated transfer of more/less rules to novel numerical quantity. In any respect, it needs to be clarified precisely why this fish training study is novel enough to warrant publication.

Previous studies in fish that have attempted to control for non-numerical variables during the learning process were very limited in the control of continuous physical variables, and we believe this is a crux in the literature because it opens the flank to criticisms such as those put forward for instance by Leibovich-Raveh and colleagues. Previous studies focused only on the overall elements’ area, the elements’ density and convex-hull (e.g. Agrillo et al., 2012, Bisazza et al., 2014, DeLong et al., 2017, now quoted in the text). Lack of control of spatial frequency and contour length in these studies is particularly unfortunate with respect to the kind of criticisms mentioned above. In our study, however, all the geometrical constrain were controlled. These non-numerical cues did not correlate with the animals’ performance accuracy (as demonstrated by the statistical analyses we added), ensuring that the discrimination made by archerfish was based on purely numerical information.

Note also that this is the first time that in fish, the possible role of spatial frequency of the stimuli used was checked for, showing that this variable is not influencing archerfish performance as well. Moreover, we controlled for several variables simultaneously, making highly unlikely the *ad hoc* argument that animals continuously shift in the use of different continuous variables.

Finally, since previous findings in vertebrate species showed the spontaneous use of relative numerical judgements only in comparative assessments of large numerosities (>4), here we provide evidence that the same rule is engaged even when comparisons involve small and large numerosity (Exp.1) or small numerosities only (Exp.2). The latter condition offered us a direct comparison with invertebrate evidence (tested only with small numbers) that was not possible before.

Reviewer #1:In the study, archerfish were trained to choose between two stimuli that each contained a certain number of dots. After a learning criterion was achieved a test phase was introduced in which it was demonstrated that fish continued to choose targets according to the relative number of dots that was rewarded at training (either the larger or smaller set). The stimuli were designed such that they controlled for continuous physical variables to ensure that those variables could not account for the preferences of the fish. The results demonstrate that fish are able to learn an abstract relative numerical rule. Although the results are convincing, it is still not clear whether those continues variables that were not controlled in any given condition of geometrical control may have influenced performance as well (as it is impossible to control for all continues variables simultaneously). In addition, during the learning phase, the only predictive variable for reward was numerosity, but on different trials different continuous physical variables were still correlated with numerosity. Although numerosity was the only consistent predictive variable, rendering it much more salient for learning, it is still possible that other continuous variables still played a role in the learning process. For instance, when radius was controlled for, a summation of the area and perimeter might still impact performance. This might have biased learning towards the use of relative numerical rule and not precise numerical representations. As a result, it seems that the authors’ claim for “spontaneous use of abstract relative numerical information in archerfish” might be somewhat misleading.

We thank the Reviewer for the comments.

To better understand the role played by continuous variables that were not controlled among the different trials, we analyzed whether the fish accuracy was changing among the different non-numerical control conditions. This was already reported in the preview version of the manuscript related to the Test phase performed both in Experiments 1 and 2. In the revised version, however, we have better clarified this part, providing further analysis, extended even to the training phase (by analyzing the performance of the two last sessions, when learning criterion was reached and confirmed). See page 8 lines 136-154, page 10 lines 182-198, pages 12-13 lines 233-246.

A more detailed explanation of our randomization logic (of the controlled physical variables across trials) was also added (page 29 lines 558-571, page 30 lines 579-584), helping to clarify how we can exclude with our analysis that fish based their choice on not controlled variables.

1) I was wondering if a similar approach to that employed by Leibovich-Raveh et al., could also be used in the current study? Can the authors examine the influence of the other continuous variables that were not controlled for in each test condition on the fish performance?

As suggested by the Reviewer, we performed a similar analysis as that employed by Leibovich-Raveh et al. The different conditions applied to the stimuli were divided into three different levels of congruity, namely the non-numerical variable that were varying congruently with numerosity (i.e. perimeter balanced: one level; area balanced: two levels; radius fixed: three levels). This analysis was performed for Experiments 1 and 2, both for the test phase as well as for the learning phase (i.e. the sessions in which the learning criterion was reached and confirmed). No improved accuracy was found among the different levels of congruity in any of the conditions analyzed.

In the main text, we have described the analysis performed and discussed the results in a dedicated paragraph (pages 15-18, lines 288-347).

2) If my first comment can be addressed, it should still be emphasized that archerfish can learn to use abstract numerical rules, but that this conclusion might be restricted to conditions in which this is the only variable that is systematically reinforced, and that this might not represent the way archerfish process numerosity in more ecological settings, in which continues variables are highly correlated with numerosity.

We agree and in the discussion we have specified that despite our study providing direct evidence of archerfish' ability to learn abstract numerical rules, this does not directly imply that number is spontaneously used in an ecological environment. Here fish were guided to use numerical cues since it was the only variable systematically reinforced. In more ecological settings, continuous variables highly correlate with numerosity, with a consequent difficulty to understand which type of information archerfish spontaneously primarily rely on and whether this could force fish to use absolute or relative magnitudes. (page 23 lines 438-445).

Also, the last sentence in the abstract was rephrased as follows: Results provide evidence that archerfish spontaneously use abstract relative numerical information for both small and large numbers when only numerical cues are available.

Reviewer #2:The authors have designed an elegant set of experiments to test the non-symbolic enumeration abilities of archerfish. These fish present a unique spitting behaviour to indicate their choice, so they are quite an ideal fish model to train and test in these types of choice behaviour. The experiments in this paper describe that archerfish generalise rules from a learned set of stimuli to a novel set. In each trial, fish are presented with two choices, each a display containing a number of black dots, of which the fish can consider and choose one. Once they learn to choose either the smaller of the two sets (containing fewer dots) or the larger, the authors use interspersed probe trials to test the basis on which the fish make their choice. The authors conclude that archerfish use the relative magnitude rule they learn during training to apply to new set of numbers.The strengths of the paper are in the complete set of stimuli used in these experiments and the choice of number sets to address the question whether fish use relative versus absolute magnitude. The authors have created a rich set of stimuli including many controls for visual features that co-vary with the number of dots, like total visual area, perimeter and the convex hull. The authors additionally do a spatial frequency analysis of the stimuli used, which is novel and adds another visual control for the stimuli.Unfortunately, the main question this manuscript addresses is familiar territory. Similar results have been reported in other species of fish, including studies by the authors. A common finding across these studies makes us expect that fish use relative magnitude rather than absolute magnitude. The authors discuss this in the manuscript as well.Another weakness in the paper is that the methods and results are missing some crucial details, not including precise numbers of trials and the effect sizes of the tests, for example.This sparsity of description makes it hard to judge whether the conclusions that the authors draw are well supported.This dataset is valuable to the community as part of a growing set of studies investigating the numerical abilities of various species and relating these to their environment and ethology.

Thank you for your comments.

We added in the main text all the information required. The method section has been integrated with further details reporting how the control of the physical variables was made and how these were specifically randomized among trials both during the training and the test phases. The statistical analysis part has been implemented as suggested by the Reviewer with further data supporting our claims.

The questions addressed in our manuscript has relevance on different points:

– In our study, we have dealt with the geometrical constraints of the elements sets taking into account all the possible combinations available. This offers a detailed approach compared to the present literature, in which the control for non-numerical variables during a learning process was only partial, mainly focusing on the overall elements’ area, the elements’ density and convex-hull. In our study, all the possible geometrical constraints were carefully addressed (radius, total area, total perimeter, mean inter-distance, convex hull and spatial frequency). Moreover, we ensured that physical cues did not alter archerfish’ performance accuracy with specific analysis.

– For the first time in fish, we focused our attention on the analysis of the spatial frequency of the stimuli used, showing that this variable is not influencing archerfish performance.

– Related to the main question of the present study, previous findings in vertebrate species showed the use of relative numerical judgements only in comparative assessments of large numerosities (>4), here we provide evidence that the same rule is engaged even when comparisons involve small and large numerosity (Exp.1) or small numerosities only (Exp.2). The latter condition offered us a direct comparison with invertebrate evidence (tested only with small numbers) that was not possible before.

All these points have been addressed and better specified in the revised version (see also comments below for a detailed description of the information integrated).

Major concerns:1. The manuscript is missing a justification for the numbers used, especially when the authors compare the numbers used to those used in other fish studies or bee studies and state that '3' belongs to a small number, perhaps can be processed by the object tracking system rather than the approximate number system (for larger numbers).

The numerosities used were justified by the aim to investigate how fish deal with discriminating numbers that could be supported by the Approximate Number System (large numbers, > 4) and the Object Tracking System (small numbers; ≤ 4). Previous studies showed the use of relative rules in fish only with large numerosities, leaving open the question of whether the same rule would be engaged even with comparisons among small numbers (see pages 5-6, lines 104-109).

2. To support the claims that fish have an ethological preference for relative versus absolute magnitude, the authors would need to show additional data, perhaps at the learning stage, of trials to criterion to learn absolute magnitude.

Please note that we were not claiming for differences in training between learning absolute and relative number discrimination but rather that, when animals have the possibility for an option, they would rely on relative rather than absolute discrimination. The logic of our task was as follows. In the training phase, the use of either absolute or relative numerical cues represented reliable strategies the animals could use to get a reward. Generalization tests with novel numerical comparisons revealed, however, that archerfish had spontaneously relied on relative rather than absolute judgements. Of course, we are not arguing that they cannot learn the latter or that it would be more difficult to learn.

In the revised version, we have extended the discussion to clarify these issues. Despite our study provides direct evidence of archerfish's ability to learn abstract numerical rules, this does not directly imply that relative number is always spontaneously used in an ecological environment. In our task, fish were in a sense "guided" to use numerical cues since it was the only variable systematically associated with reinforcement (others, continuous variables were not). In more ecological settings, however, continuous variables highly correlate with numerosity, with a consequent difficulty at disentangling which type of information archerfish spontaneously primarily rely on (see page 23 lines 438-445).

3. The description of the methods and results are lacking in a level of detail necessary to evaluate the manuscript, some of which I list below:4. The average number of trials that each fish performed are never mentioned, but since there are only 24 test trials, 8 presented per day for 3 testing days, some fish might not see all the stimuli. Across 3 tests in Experiment 1, fish would have to see 6 control images for each number set. I'm not sure how that is accomplished.

All the missing information have been added in the revised version. We specified in detail the number of trials for each control condition both for the training and test phase, justifying the numbers used. See page 29, lines 558-571 and page 30, lines 579-584.

5. The bar plots in Figures 2 and 3 should plot each individual fish and describe what the error bars describe.

We uploaded new graphs plotting each individual fish for Figures 2 and 3 (now Figures 3 and 5, respectively). Error bars report the Standard Error of the Mean (SEM), as specified in each Figure caption.

6. Effect sizes need to be reported alongside the p-values, for the GLMMs as well as the other analyses.

Log odds ratios from the best fits are now reported as GLMM estimates with their errors and also converted in natural scales to give a clearer interpretation of the effect size (which for our binomial distributions corresponds to the sample proportion). 95% confidence intervals were also reported for a cleaner interpretation of the final results. Moreover, following Cohen 1988, the chance proportion of 0.5 was subtracted to our binomial sample proportion to obtain a final Cohen’s g effect size (interpretable as: <0.05 negligible, 0.1-0.15 small, 0.2-0.25 medium, >0.25 large).

7. Figure 4 contains no description of what the individual dots on the right side correspond to. There are 6 dots in each panel.

We uploaded a new Figure (now Figure 7) with the required information.

8. Were the control sets randomized or were they changed by session. Figure 1 gives the impression that they are deployed in individual test sessions.

In every session, all the controls sets were randomized in every session. A detailed description of the control condition and of their randomization can be found at lines page 29, lines 558-571 and page 30, lines 579-584.

9. Were the fish assigned to num 3 and num 6 training randomly? Or was this based on some shaping criteria? Additionally, please describe the criteria used to exclude the 4 fish. How many no-choice trials?

Fish were assigned to each condition randomly (page 25, line 482). A specific description related to the animals excluded has been reported; fish were excluded because they did not show any consistent motivation in hitting the screen, failing to get through the different steps of the pre-training phase. See page 25, lines 482-485 and page 28, lines 538-542.

Reviewer #3:The manuscript is devoted to studying archerfish number discrimination. The authors rely on their own generated computer code to generate stimuli for number discrimination task. They control some visual variables that are usually correlated with numbers of objects. The study is interesting and contributes to the growing evidence on numerical capabilities in branches of evolution far from mammals.1. The evidence that non-numerical variables cannot influence the fish decision is light. The evidence relies on a computer code presented elsewhere. While I am fully aware that controlled everything is hard (or even impossible), I suggest that the author will make efforts to convince the reader that some visual variables were controlled and explain which variables were not controlled.

In the revised version, we have provided a detailed explanation of how the stimuli were controlled and which are the variables that cannot be controlled for each condition. Given the geometrical limitations, we stressed how the use of different randomized control conditions allows making none of the physical variables reliable in the task, differently than numerical information (see page 27, lines 513-526).

In addition, a novel specific paragraph aimed to investigate whether non-numerical magnitudes influenced archerfish performance accuracy shows more in detail which are the variables controlled and not controlled in each condition (pages 15-18, lines 288-347).

2. The data is reported on the entire population level. I think that single fish data is critical to assess the results. The success rate reported might be due to single fish which maintains high performance.

New graphs have been reported in the revised version, showing the average and each individual performance (see Figures 3 and 5). Moreover, Individual performance data for each non-numerical control condition in the training and the test conditions are made available in Supplementary Materials.

3. I assume that not all fish succeeded in the task and the reported fish are only fish which managed to achieve a criterion. A report on this part of the study will be useful.

Four archerfish were excluded from the study because they did not show any consistent motivation in hitting the screen, failing to get through the different steps of the pre-training phase. See lines 482-485 and 538-542 for a detailed description.

4. The authors have data on the learning part, I suggest presenting and analyzing it.

Concerning the learning part, for both Experiments 1 and 2, we have reported graphs with the learning curves of each individual. Moreover, we conducted specific analysis on the two last learning sessions, when the criterion was reached and confirmed, to investigate further whether the performance accuracy was influenced by specific physical variables and not only numerical cues. See pages7-8, lines 125-154 (Experiment 1) and page 12-13, lines 231-246 (Experiment 2) and the Supplementary Materials.

5. Method training phase: the experimental procedure is identical to Ben-Tov 2015, this should be mentioned.

We quoted the study by Ben-Tov and colleagues. Thank you.

6. Given lines 222-225: what is new in the current study?

Lines 222-225: “Considering the results of Experiment 2 and Experiment 1, it is apparent that archerfish can easily discriminate between small and large numerosity using the same rules, providing evidence in favour of a unique system underlying numerical 224 discrimination as found in other fish species [26,58].”

From this study, we have demonstrated that archerfish performance in discriminating between numerosities was not affected by the presence of small or large numbers, as hypothesized belonging to the “Object Tracking System” and the “Approximate Number System”, respectively. Our results support the hypothesis of a unique approximate number system for representing numerosities in archerfish, working both for small and large numbers. This evidence is in agreement with other fish studies that support the same idea but use different methods (e.g. spontaneous choice procedures).

Besides this, what is new in the current study is the use of accurate control of the physical variables in such a way that has never been done before in fish. Previous studies were all incomplete as to the control of continuous physical variables, focusing only on the control of some non-numerical variables such as overall elements’ area, density and sparsity (convex-hull), with no control on other crucial features such as the perimeter, the individual elements’ size and the spatial frequency. Note also that differently than in previous study we controlled for several variables simultaneously. Thus here we have demonstrated for the first time in archerfish the ability to learn a discrimination based only on a numerical rule, excluding the possible influence of all other non-numerical variables.

The second novel evidence from the current study is the use of a relative numerical rule by archerfish in numerical comparisons involving small and large numerosity (Exp.1) or small numerosities only (Exp.2). Previous evidence in vertebrate species showed relative numerical judgements only in comparative assessments of large numerosities (>4). Moreover, our study offered us a direct comparison with invertebrate evidence (tested only with small numbers) that was not possible before.

Lastly, as previously reported, for the first time in fish, we focused our attention on the analysis of the spatial frequency of the stimuli used, showing that also this crucial continuous variable is not influencing archerfish performance.

7. Discussion: As the author acknowledged, Leibovich-Raveh et al., showed that when archerfish make magnitude-related decisions, their choice is influenced by the non-numerical variables that positively correlate with numerosity. A clear statement why the current study is different is needed (see comment 1 above).

In the revised version, we addressed this question (see paragraph Accuracy is not influenced by non-numerical magnitudes, pages 15-18, lines 288-347). We have analyzed our data using a similar approach as that employed by Leibovich-Raveh et al., in order to investigate whether the choice was influenced by the non-numerical variables that positively correlate with numerosity. We also discussed the difference between our study and that of Leibovich-Raveh and colleagues (lines 339-347).